

# Both landscape heterogeneity and configuration determine Woodlarks (*Lullula arborea*) breeding territories

Marlies Resch and Marcela Suarez-Rubio

Institute of Zoology, Department of Integrative Biology and Biodiversity Research, University of Natural Resources and Life Sciences, Vienna, Austria

## ABSTRACT

Farmland birds have declined in the last decades mostly due to agriculture intensification. The Woodlark *Lullula arborea*, a farmland species of conservation concern and protected by the European Bird Directive, occurs in a variety of habitats across its geographic range. Although habitat heterogeneity has been recognized as a key feature, the preference or avoidance of particular habitat attributes might differ across its range because different localities may have distinct conditions. Such variation would challenge conservation efforts at the local level. Our aim was to assess habitat associations of Woodlarks and determine whether the habitat attributes identified as important in other locations across its range could be generalised and applied to Austrian populations. In addition, habitat associations can be influenced by land-use change. We examined changes in land use from 2007 to 2016 in 15 municipalities surrounding areas occupied by Woodlarks. We quantified the composition and configuration of the local landscape surrounding 18 singing males' territories and 16 non-territory sites. We found that the probability of Woodlarks territories increased with landscape heterogeneity between 50% and 70%, increased with dispersed bare soil patches, decreased with overall patch density and were away from dirt roads. Contrary to our expectation, there was no indication of land-use change. In contrast to previous studies, vegetation height, the presence and proximity to woodland were not identified as important habitat characteristics. Thus, some conservation recommendations can be derived from other localities, for example, maintaining or enhancing landscape heterogeneity. However, others should be adapted to local conditions. In Austria, conservation efforts should focus on including dispersed patches of bare soil and limiting the development of dirt roads nearby Woodlark territories, in addition to promoting a heterogeneous landscape.

# INTRODUCTION

Farmland birds are declining at unprecedented rates in Europe, although the rate of decline may have decreased since the mid-1990s (*e.g.*, *Donald et al., 2006*; *Gregory et al., 2019*). An important driver for this decline is agricultural intensification

Corresponding author
Marcela Suarez-Rubio,
marcela.suarezrubio@boku.ac.at

(*Chamberlain et al., 2000*; *Jerrentrup et al., 2017*), which includes changing the crop management including type and relative abundance of different crops, shifting the timing of some agricultural management activities and increasing the use of artificial fertilizers and herbicides (*Chamberlain et al., 2000*; *Donald, Green & Heath, 2001*; *Stanton, Morrissey & Clark, 2018*). In addition, habitat homogenisation through the loss or reduction of important landscape elements like hedgerows has also contributed to the decline of farmland birds (*Benton, Vickery & Wilson, 2003*; *Stanton, Morrissey & Clark, 2018*). Intense farming can modify the preferred farmland bird habitats which affect the required conditions for both breeding habitat and food resources (*Gil-Tena et al., 2015*). For farmland birds, therefore, detailed knowledge of the habitat requirements of each species is key for augmenting their populations (*Whittingham et al., 2005*).

One of the farmland birds susceptible to habitat modification caused by agricultural intensification is the Woodlark (*Lullula arborea*). Woodlarks are insectivorous and ground-nesting birds that occur mainly in Europe, northern Africa and western Asia. They are listed in the Annex I of the European Bird Directive (79/409/EEC of 2 April 1979). Although there has been a recent trend toward a moderate increase (*EBCC, 2021*), the population size has fluctuated greatly in Europe recently, mainly due to habitat changes on their breeding grounds (*Takacs et al., 2020*). In Austria, Woodlarks are classified as vulnerable in the Austrian Red List (*Dvorak et al., 2017*), and few populations occur in Upper Austria (*Uhl & Wichmann, 2013*), Lower Austria (*Straka, 2008*) and Lake Neusiedl (*Dvorak et al., 2009*). In Upper Austria, the Woodlark population has decreased from 38–42 breeding territories in 2007 to 16–18 breeding territories in 2017 (*Uhl & Wichmann, 2018*). Thus, identifying local habitat associations and understanding the role of local land-use change in altering their habitats is of great importance.

It is common that bird-habitat associations measured in one area are taken as representative for the species, like a species-specific trait (*Wesołowski & Fuller, 2012*). However, habitat associations are not temporally/spatially uniform for many bird species (*Fuller, 2002*; *Fuller & Rothery, 2013*). These associations can change due to, for example, changes in the type of habitat available, behavioral flexibility, or variations in population density (*Havlíček et al., 2021*; *Newson et al., 2009*; *Wesołowski & Fuller, 2012*). In Britain, for example, Woodlarks habitat associations have changed over time (*Wright et al., 2007*). They have been traditionally associated with heathland (*Holloway, 1996*). However, since the 1970s, they are associated with clear-fell forestry habitats (*Sitters et al., 1996*; *Wotton & Gillings, 2000*). This shift has occurred due to changes in land use and habitat availability (*Holloway, 1996*; *Sharrock, 1976*), which highlights the importance of evaluating land-use change as a potential factor affecting habitat associations.

In addition, bird-habitat associations can vary spatially across a species' geographic range. Woodlarks, for example, are associated with heathlands (*Mallord et al., 2007a*) and forest clear cuts in Britain (*Wright et al., 2007*), Christmas-tree plantations in Germany (*Fartmann et al., 2018*), vineyards in Switzerland (*Arlettaz et al., 2012*; *Bosco, Arlettaz & Jacot, 2019*; *Buehler et al., 2017*), and crop-steppes in Italy (*Campedelli et al., 2015*).

Therefore, a species might occupy different habitats across its range to reflect different limiting factors or abiotic conditions (*e.g.*, *Boves et al., 2013*; *Koleček, Reif & Weidinger, 2015*; *Piotr et al., 2011*; *Wesołowski & Fuller, 2012*; *Whittingham et al., 2007*).

The preference or avoidance of habitat attributes has been evaluated previously in different Woodlark habitats and at different scales in an attempt to understand population trends and develop conservation strategies. It has been found that Woodlarks are associated with tall and dense ground vegetation (*Buehler et al., 2017*; *Mallord et al., 2007a*) at the microhabitat scale (~5 m around the nest) in vineyards and heathlands. Further, a mosaic of grass, herbs and bare soil is preferred in orchards and vineyards (*Arlettaz et al., 2012*; *Schaub et al., 2010*). Besides bare soil, open grassland and sparse cover of bushes and trees are favoured at the mesohabitat scale (~50 m) in grasslands/croplands, heathlands and forestry areas (*Sirami, Brotons & Martin, 2011*; *Sitters et al., 1996*). An interplay between habitat amount (*i.e.*, area of available habitat) and habitat fragmentation emerges in vineyards at the macrohabitat scale (~100 m) (*Bosco et al., 2021*). If habitat amount is below 20% Woodlarks avoid fragmented areas, but if habitat amount exceeds 20% then there is a preference for fragmented areas. In addition, the degree of connectivity is also relevant when the spatial arrangement of habitat elements is evaluated in crop-steppes at the macrohabitat scale (*Campedelli et al., 2015*).

Although habitat heterogeneity at multiple scales has been identified as a key factor for Woodlarks (*Sirami, Brotons & Martin, 2011*) and farmland birds in general (*Benton, Vickery & Wilson, 2003*), it is unclear whether other previously identified habitat associations can be generalised to other habitats. If habitat preferences recognised in one habitat do not apply in other habitats, then it may result in developing inappropriate conservation strategies. Indeed, effective conservation measures depend on detailed knowledge about the variation in response across the species' range. Hence, management strategies must also vary locally (*Wesołowski & Fuller, 2012*; *Whittingham et al., 2007*).

The aim of this study was to assess habitat associations of Woodlarks in a cropland-grassland-forest mosaic in Upper Austria and determine whether the habitat characteristics identified as important in other habitats across its range (*e.g.*, vineyards) were also important or have similar ranking of importance. We evaluated both habitat amount and the spatial arrangement of habitat elements and human features (*i.e.*, configuration) at the macrohabitat scale (*i.e.*, local landscape *sensu Fahrig, 2013*) to better understand whether previous conservation recommendations can be applied in other habitats or should be verified locally for effective conservation measures. In addition, we examined changes in land use from 2007 to 2016 because ongoing agriculture intensification, in particular changes in type and area occupied by different crops, has been documented in Upper Austria (*van der Sluis et al., 2016*). Additionally, change in areas of major land-cover types (*e.g.*, grassland) could result in simple reduction of the amount of habitat available for Woodlarks (*Reif & Hanzelka, 2016*). Altogether, this knowledge can be used to provide adequate support to this vulnerable species.

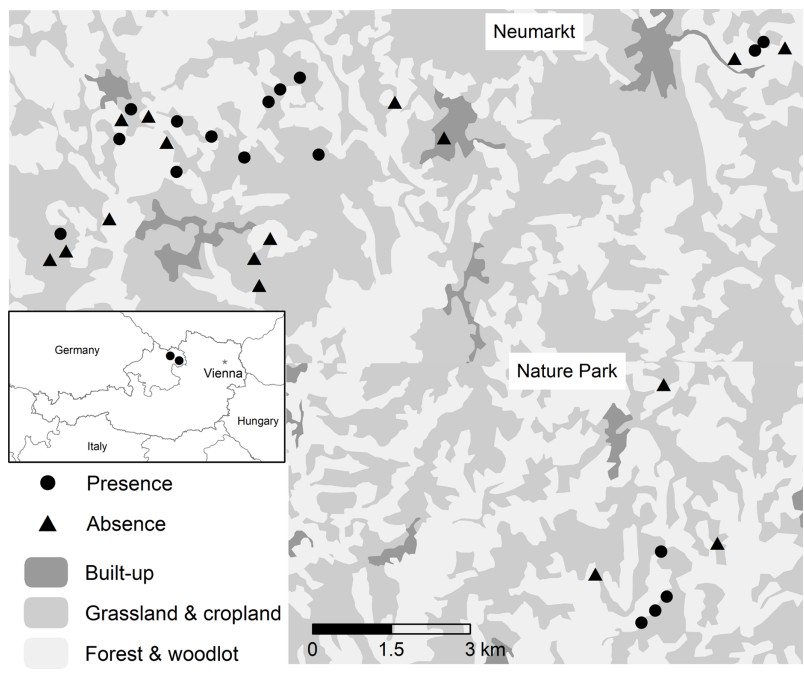

**Figure 1 Location of the field sites in Neumarkt and Nature Park in Upper Austria, Austria.**
Occupied Woodlark territories are depicted with circles ($n$ = 18) and non-territories with triangles
($n$ = 16). Land cover based on the 2018 CORINE Land cover map available at the Copernicus Land
Monitoring Service (https://land.copernicus.eu/).

# MATERIALS & METHODS

## Study area

The study was conducted in the Mühlviertel region located in the north-eastern part of
Upper Austria, Austria (Central Europe) (Fig. 1). It covers an area of 3,090 km$^2$ and
has around 270,000 inhabitants. Within the Mühlviertel region, we focused on the Nature
Park (Rechberg 48°19′N, 14°42′E) in the east and Neumarkt (Neumarkt im Mühlkreis
48°25′N, 14°29′E) in the north. These are areas located approximately 20 km apart
and include the main population of Woodlarks in Upper Austria (*Uhl, 2009*; *Uhl &
Wichmann, 2013*). We received oral consent from Barbara Derntl from the Nature Park
Mühlviertel to access their premises. Some sampling was conducted on private land with
the owners' consent.

The region has a continental climate, and the mean annual temperature is between
5–9 °C. The area is characterized by hills and a mixture of forest, grassland and cropland.
In some parts, the region is rich in habitat elements, like groves (clusters of trees), isolated
trees and hedgerows, and in other parts these elements are scarce. Around a quarter of
the agricultural area is cultivated organically, and the average farm size is around 30 ha
(*BMLFUW, 2017*).

## Territory mapping

Based on previous sightings and territories (*Uhl, 2009*; *Uhl & Wichmann, 2013*), we
mapped territories of 18 singing males during the breeding season of 2017 to determine the

distribution of territorial Woodlarks throughout the study area (Fig. 1). The territory mapping was performed following *Südbeck et al. (2005)*. The area was surveyed six times systematically from 13 March to 19 May between sunrise and 10:00 h during days without rain or strong winds (Beaufort wind force < 4). During each survey, we recorded Woodlark's location using a global positioning system, behaviour (singing or foraging) and position (*e.g.*, on the ground, on top of tree). Observations of individuals singing in flight were excluded as they could not be associated with any habitat use. The territory centre either corresponded to the centre of the Woodlark territory in most cases or to the nest, in the few instances where we were able to find the nest. Using the function "create random points" in ArcGIS v. 10.5.1 (*ESRI, 2017*), we randomly selected 16 locations within the study area where no Woodlarks were recorded and corresponded to pseudo-absence (hereafter referred as "absence"). Random points that fell on water bodies or on already occupied territories were excluded. The minimum distance between Woodlark territories and absence areas was 266 m (1,150 ± 718 m) and there was little or no overlap between territories.

## Habitat characteristics

We established a 150 m radius around the centre of the territories and absence areas, which equals approximately 7 ha. We selected this size to capture the size range of Woodlarks territories (*Harrison & Forster, 1959*). It also corresponds to the average territory size of Woodlarks in this region (*Uhl, 2009*). For characterizing the macrohabitat within the study plots of 7 ha between April and May 2017, we divided habitat elements into two categories: land use/land cover and linear elements. We assigned 11 land use/land cover classes: bare soil, grassland, cropland with short (<20 cm) and tall (20–150 cm) vegetation, rough pastures (*i.e.*, non-intensive grazing pastures), forest, groves, residential areas, dirt roads, asphalt roads, and water bodies. We estimated the area covered for each of the land use/land cover classes, measured the height of grassland, cropland, rough pastures, vegetation between the track lanes of dirt roads, estimated the height of forest, groves, and measured the diameter at breast height (DBH) for forest and groves. Given that vegetation patches varied in size, we allocated a number of measuring points in each vegetation patch based on its size (*e.g.*, from two measuring points in patches of <0.7 ha up to 20 measuring points for patches of >6.4 ha). The minimum distance between measuring points was 10 m and the placement of the measuring points was random. Land use/land cover patches smaller than 15 m$^2$ were not characterised and were included as part of adjacent larger patches. In addition, we measured the length of linear elements such as dirt roads, asphalt roads, electricity lines, and hedgerows. We estimated the distance from the centre of the territory to linear elements and to forest and groves.

We used ArcGIS v. 10.5.1 (*ESRI, 2017*) to digitise the collected field data. To determine the arrangement of the different land use/land cover classes in the sampling plots (*i.e.*, spatial configuration) we calculated landscape metrics using FRAGSTATS v 4.2 (*McGarigal, Cushman & Ene, 2012*). For the whole sampling plot, we calculated overall patch density (number of patches per 100 ha) as it is a useful metric of landscape

configuration in which it indicates whether patches were small and numerous in the landscape or if they were mainly large and few patches. Landscape shape index measures the overall geometric complexity and can be interpreted as a measure of landscape dispersion—the greater the value, the more dispersed are the patch types (*McGarigal, Cushman & Ene, 2012*). Mean proximity index calculates the degree of patch isolation by considering both the size and the proximity to all patches. The index distinguishes sparse distributions of small habitat patches from configurations where the habitat forms a complex cluster of larger patches. Contagion index measures both patch type interspersion (*i.e.*, the intermixing of different patch types) as well as patch dispersion (*i.e.*, the spatial distribution of a patch type), and Simpson's diversity index calculates the heterogeneity of the landscape (*McGarigal, Cushman & Ene, 2012*). A higher value of Simpson's diversity index means greater compositional heterogeneity. We also calculated patch density and the landscape shape index for each of the land use/land cover classes (Table S1).

## Land-use change

We used agricultural land-use data of the area of Mühlviertel from the years 2007, 2012 and 2016 provided by the Austrian Federal Ministry of Agriculture, Forestry, Environment and Water Management to identify land-use changes. The focus of this analysis was on the 15 municipalities where Woodlarks were recorded from 2007 until 2016 (*Uhl, 2009*; *Uhl & Wichmann, 2013*, *2018*). For the 15 municipalities in this region, four land-use types were analysed: Grassland included wildflower margins, permanent pasture, managed meadows (cut one to three times per year) and seeded pastures. Cropland included different types of legumes, field forage, summer grain, winter grain, potato, corn and other field crops. Woody vegetation referred to Christmas trees, energy forests (*i.e.*, fast-growing trees with the aim of producing wood chips), tree nurseries, and different types of fruit trees. Protected comprised protected arable land ('*Landschaftselement Acker*'), protected grassland ('*Landschaftselement Grünland*'), protected natural monuments and protected areas that have good agricultural and environmental conditions (GAECs).

## Data analysis

We performed a conditional Random forest algorithm (*Breiman, 2001*; *Hothorn et al., 2006*) to rank the 56 explanatory variables (Table S1) according to their importance. The magnitude of importance of the predictors was compared using the Conditional Variable Importance values from the random forest approach. Conditional Variable Importance calculates the mean decrease of prediction accuracy of the response variable devoted to an explanatory variable after permuting it over all data and avoids overestimating the importance of correlated predictor variables (*Strobl et al., 2008*). We used the cforest function from the R package "partykit" (*Hothorn & Zeileis, 2015*) with 5,000 bootstrap samples and mtry = p/3 variables at each split.

We checked for multicollinearity of the most important variables identified by the conditional Random forest algorithm using variance inflation factor (VIF) with the R-package "usdm" (*Naimi, 2015*). Those variables with VIF > 2 were excluded from further

analysis. To determine which habitat characteristics were the most important for the Woodlarks in the Mühlviertel, a Generalised Estimating Equation model was performed with the response variable absence (0) and presence (1) of Woodlarks in the study plots and the remaining five explanatory variables. These variables were landscape heterogeneity, patch density, landscape shape index of bare soil, percentage of dirt roads, and distance from dirt roads. We included the region as a random factor and the correlation structure "AR-1" to account for the spatial correlation of the data (package "geepack"; *Højsgaard, Halekoh & Yan, 2006*). Model selection was completed *via* model averaging (package "MuMIn"; *Barton, 2020*) to show the influence of all variables where QIC (Quasi Information Criteria) change was smaller than two (*Zuur et al., 2009*).

We performed a compositional data analysis to test whether land-use types (*i.e.*, grassland, cropland, woody vegetation and protected) changed in the 15 municipalities where Woodlarks occurred from 2007 to 2016. The response variable was the land-use type and represents compositional data because scores for each class are proportions of the total area covered and therefore are interdependent (*Aitchison, 1982*). The explanatory variable was year (2007, 2012 and 2016). We performed an analysis of variance (ANOVA) adjusted to compositions (*van den Boogaart & Tolosana-Delgado, 2013*) as this technique accounts for the dependence of the compositions and inspected the residuals and checked for symmetry and normality within the package "compositions" (*van den Boogaart, Tolosana-Delgado & Bren, 2021*). All the statistical analyses were done with R v. 4.0.3 (*R Core Team, 2020*).

## RESULTS

The most important variables for the occurrence of Woodlark territories were landscape heterogeneity, length of dirt road, proportion of dirt roads, overall patch density, landscape shape index of bare soil, patch density of grassland, proportion of bare soil, distance from dirt roads, and contagion index (Fig. 2). The variance inflation factor (VIF) of the most important variables showed that length of dirt road, patch density of grassland, proportion of bare soil, and contagion index had a VIF > 2, and they were therefore excluded from further analysis.

The Generalised Estimated Equation model showed that the strongest predictors on the occurrence of Woodlark territories were landscape heterogeneity, distance from dirt roads, landscape shape index of bare soil and overall patch density (Table 1). All Woodlark territories occurred in areas with a mixture of grasslands (average 25%), croplands with short (<20 cm) and tall (20–150 cm) vegetation (12% and 21%, respectively), forest (23%) and bare soil (10%). The probability of the occurrence of a Woodlarks territory increased sharply with landscape heterogeneity between 50% and 70%, increased with the degree of dispersion of bare soil patches, increased with distance from dirt roads, and decreased with overall patch density (Fig. 3).

When evaluating land-use types in the 15 municipalities where Woodlarks occurred from 2007 to 2016, cropland covered most of the area (66.6%), followed by grassland (28.9%). Woody vegetation and protected areas together covered 4.5%. The proportions of

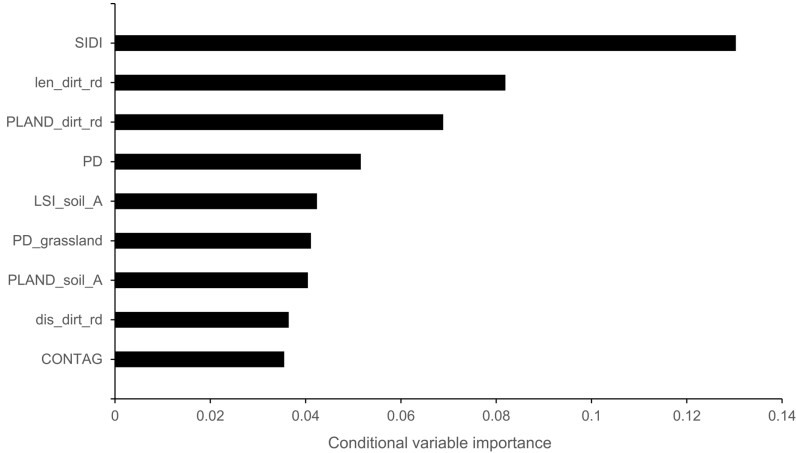

**Figure 2** **Conditional variable importance of the top nine variables based on the conditional random forest analysis.** Landscape heterogeneity (SIDI), length of dirt road (len_dirt_rd), proportion of dirt roads (PLAND_ dirt_rd), patch density of the landscape (PD), landscape shape index of bare soil (LSI_soil_A), patch density of grassland (PD_grassland), proportion of bare soil (PLAND_soil_A), distance from dirt roads (dis_ dirt_rd), and contagion index (CONTAG).

**Table 1** **Result of the Generalised Estimating Equation model with absence and presence as response variable and non-collinear predictors identified by the conditional random forest algorithm as explanatory variables.**

|  | Estimate | SE | Wald | P |
|---|---|---|---|---|
| (Intercept) | −0.416 | 0.061 | 46.9 | **<0.001** |
| Landscape heterogeneity | 1.988 | 0.287 | 47.9 | **<0.001** |
| Distance from dirt roads | 0.870 | 0.136 | 41.6 | **<0.001** |
| Landscape shape index of bare soil | 0.438 | 0.076 | 33.5 | **<0.001** |
| Patch density | −0.354 | 0.043 | 69.2 | **<0.001** |

**Note:**
The table shows the estimate, standard error (SE), Wald value and *p*-value (P), significant results are in bold.

these land-use types did not differ among years ($F_{2,21}$ = 0.8108, df = 2, *p* = 0.458; Fig. 4). Thus, there was no indication of land-use change between 2007, 2012 and 2016.

## DISCUSSION

Our results show that Woodlarks were associated with landscape heterogeneity (quantified as Simpson's diversity index), overall patch density, landscape shape index of bare soil and distance from dirt roads. Landscape heterogeneity has been previously identified as a key characteristic in other habitats across the Woodlark's range such as in Christmas-tree plantations (*Fartmann et al., 2018*), Mediterranean landscapes (*Sirami, Brotons & Martin, 2011*), low-intensity agricultural systems (*Brambilla, Falco & Negri, 2012*) and vegetated vineyards (*Bosco, Arlettaz & Jacot, 2019*). Although it was the most important predictor of Woodlark occurrence, the components of heterogeneity varied in the different habitats. For example, in Christmas-tree plantations, Woodlarks favour the high habitat structure of trees of different age growing along large areas of bare soil and gravel

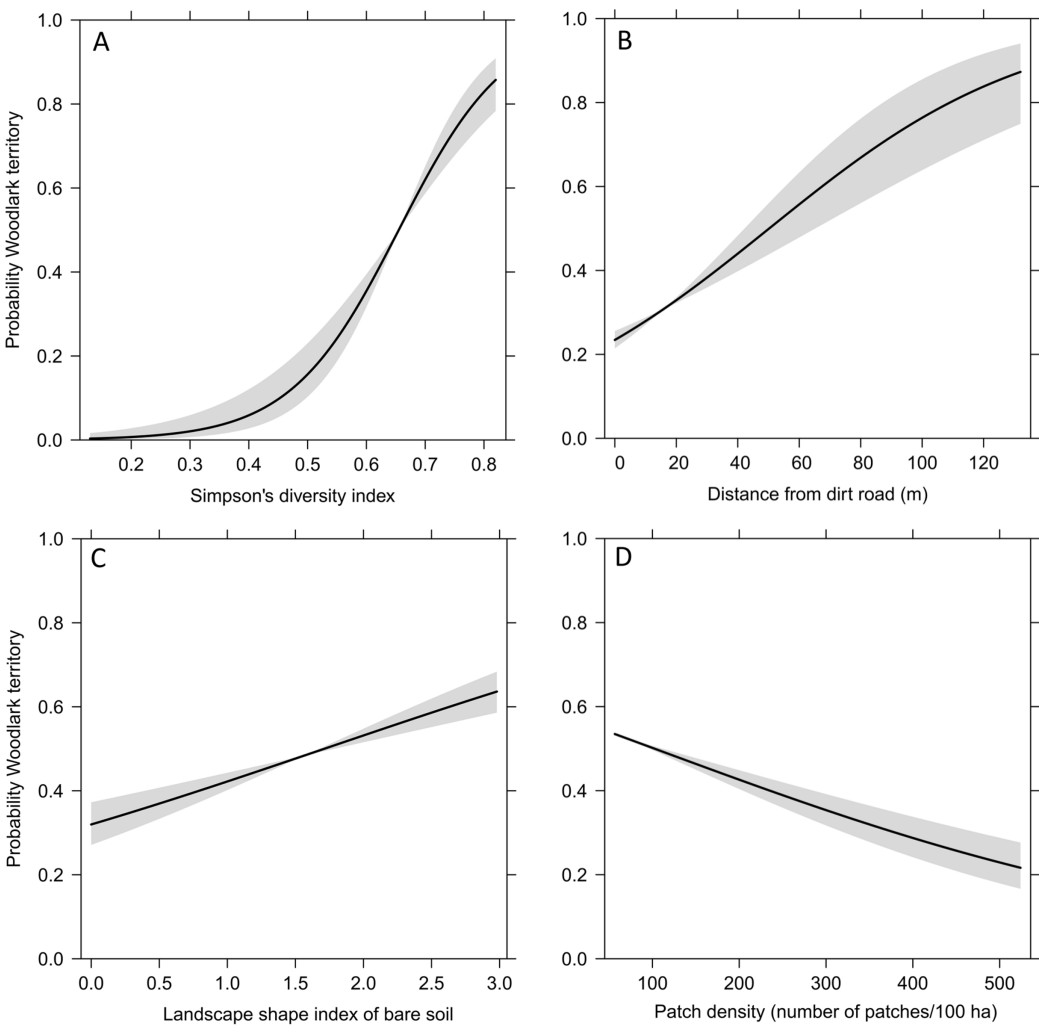

**Figure 3 Fitted values (line) and 95% confidence intervals (grey) obtained by the Generalised Estimating Equation model depicting the probability of the occurrence of Woodlark territories and significant predictors.** (A) landscape heterogeneity represented by the Simpson's diversity index, (B) distance from dirt roads, (C) landscape shape index of bare soil, and (D) patch density.

between the tree lines (*Fartmann et al., 2018*). In Mediterranean landscapes, open shrubland intermixed with tall grass/herbs and bare soil were preferred. In vineyards, Woodlarks were associated with high ground vegetation cover, plant species richness and wider inter-rows (*Bosco, Arlettaz & Jacot, 2019*). In our study, Woodlark territories were in areas with a mixture of grasslands, croplands with short and tall vegetation, forest and bare soil. Therefore, the combination of different habitat elements enhances the access to a range of resources necessary to meet vital needs as postulated in the complementation hypothesis (*Dunning, Danielson & Pulliam, 1992*). It has been shown that heterogenous landscapes offer abundant and accessible food resources for both nesting and foraging and also provide suitable cover and/or protection from predators or
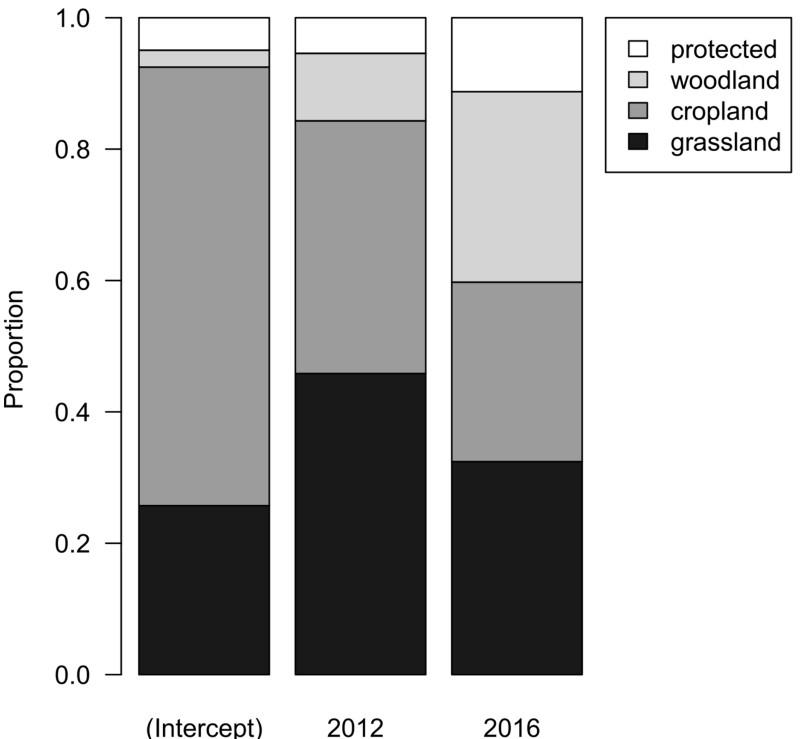

**Figure 4** **Bar chart representing the parameters of a linear model with compositional response and year as main effect.** Grey tones depict the associated land-use type: grassland (black), cropland (mid-grey), woodland (light grey), and protected (white).

harsh weather (*Benton, Vickery & Wilson, 2003*; *Lima & Dill, 1990*; *Vickery & Arlettaz, 2012*).

Interestingly, a common feature that describes Woodlarks occurrence in most studies is the presence of bare soil. The presence of few patches of bare soil have been shown to increase the attractiveness of potential breeding sites in vineyards (*Arlettaz et al., 2012*). Ground foraging insectivorous birds, like the Woodlark, forage in bare soil due to the high invertebrate prey accessibility, where they can detect and pick up prey items easily (*Schaub et al., 2010*). Rather than occurrence of bare soil patches, in our study, the arrangement—which has not been evaluated previously—was a predictor of Woodlark territories. A high degree of dispersed bare soil patches indicates the importance of bare soil scattered between grassland and cropland. Thus, the arrangement of bare soil contributes to the heterogeneity of the landscape and complements the resources found in contrasting habitats (*i.e.*, grasslands) (*Dolman, 2012*; *Pino et al., 2000*).

In our study area, bare soil was also found in the track lanes of dirt roads. However, the probability of Woodlark territories was higher away from dirt roads. This indicates possible avoidance of human disturbance (*Mackowicz, 1970*; *Rösch, Aloisio & Entling, 2021*). However, Woodlarks were seen dust bathing in the sand of dirt roads during the study period. Some farmland species might tolerate low road traffic and human presence levels if there is optimal habitat for them (*Tarjuelo et al., 2020*), which might be also the case for Woodlarks. Dirt roads could provide access to food resources on the bare

soil or in the short and sparse vegetation found between the track lanes (*Harrison & Forster, 1959*; *Schaub et al., 2010*). In vineyards, Woodlarks prefer vine-rows with a vegetation cover as these provide high abundance of invertebrate prey (*Bosco, Arlettaz & Jacot, 2019*; *Rösch, Aloisio & Entling, 2021*). Therefore, it might be a trade-off between accessing a certain resource (in this case prey) and the perceived risk posed by humans (*Mallord et al., 2007b*).

In addition, the degree of fragmentation was also relevant for the occurrence of Woodlarks territories. Territories were established in areas with lower patch density suggesting that large and few habitat patches (*i.e.*, low fragmentation) were more attractive than small and numerous patches in the local landscapes. Similarly, Woodlarks avoided fragmented areas in vineyards when the amount of habitat was less than 20% (*Bosco et al., 2021*). However, at the broad scale (1 km$^2$) Woodlarks were more abundant in fragmented steppe habitats (*Campedelli et al., 2015*). This highlights the interplay between the composition and configuration of habitat patches and the difficulty of generalising conservation strategies derived from analysis at different spatial scales.

The scale of analysis might also explain why some previously identified characteristics did not play a role in our study. Here, we focused on the local landscape. Those studies, where the height of grass/herbs was relevant, were at the microhabitat scale (immediate surroundings around the nest) (*Buehler et al., 2017*; *Harrison & Forster, 1959*; *Mallord et al., 2007a*). Alternatively, Woodlarks' habitat association might change within the breeding season (*Brambilla & Rubolini, 2009*). In our study, we focused on the first clutch, so assessment on whether habitat associations change and which factors may become an important predictor late in the breeding season requires further study. Other studies have found that Woodlarks were associated with the presence and proximity to woodland in steppe landscapes (*Campedelli et al., 2015*; *Schaefer & Vogel, 2000*) which was also not the case in our study. Although the proportion of woodland was a habitat element in our study area, it was not relevant by itself but contributed to the overall heterogeneity of the landscape. Thus, habitat associations vary across the Woodlark's geographical range as these are context-dependent (*Whittingham et al., 2007*), but landscape heterogeneity was the ubiquitous attribute at multiple spatial scales and across its range.

Interestingly, we did not detect any significant land-use changes from 2007 to 2016. The proportion of land-use types considered was similar during this period, which could be attributed to the 'stabilisation of intensification' that happened in most regions in Europe from 2001 to 2011 (*van der Sluis et al., 2016*). However, it is important to note that we evaluated these changes at the regional level and other forms of intensification like increasing the use of artificial pesticides might play a greater role at the local level (*Kirchner, Schönhart & Schmid, 2016*) and potentially affect Woodlark populations (*Kristensen et al., 2016*).

## CONCLUSION

Landscape heterogeneity was a key habitat characteristic for Woodlarks as was previously identified across its range. In addition, the configuration of habitat elements should be

considered when assessing habitat associations. Even though there was no evidence of changes in land use up to 2016, further monitoring is recommended to mitigate potential effects it might have on Woodlarks' habitats. Although some conservation recommendations can be derived from other regions, for example, maintaining or enhancing landscape heterogeneity (*e.g.*, *Bosco, Arlettaz & Jacot, 2019*; *Fartmann et al., 2018*; *Sirami, Brotons & Martin, 2011*), others should change with local conditions because important habitat characteristics vary across its range. In Upper Austria, management and conservation efforts should focus on maintaining or enhancing a mixed-habitat landscape of grassland, cropland, forest, and bare soil. These elements should be aggregated except for bare soil which should be dispersed. Finally, new dirt road development should be limited or located away from areas known to have Woodlarks territories. Together, these measures will benefit Woodlarks and inform future conservation management in Upper Austria.

## ACKNOWLEDGEMENTS
We would like to thank H. Uhl, A. Schmalzer, H. Rubenser and H. Kurz from BirdLife Austria for their support during the field mapping.

### Funding
The authors received no funding for this work.

### Competing Interests
The authors declare that they have no competing interests.

### Author Contributions
- Marlies Resch performed the experiments, analysed the data, prepared figures and/or tables, authored or reviewed drafts of the paper, and approved the final draft.
- Marcela Suarez-Rubio conceived and designed the experiments, analysed the data, prepared figures and/or tables, authored or reviewed drafts of the paper, and approved the final draft.

### Field Study Permissions
The following information was supplied relating to field study approvals (*i.e.*, approving body and any reference numbers):

We received oral consent from Barbara Derntl from the Nature Park Mühlviertel to access the premises. Some sampling was conducted on private land with the owners' consent.

### Data Availability
The raw data for performing the analysis about the habitat associations of the Woodlark are available in the Supplemental Files.

## Supplemental Information

Supplemental information for this article can be found online at http://dx.doi.org/10.7717/peerj.12476#supplemental-information.

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
