# Peer review of "Both landscape heterogeneity and configuration determine Woodlarks (Lullula arborea) breeding territories"

_PeerJ, doi:10.7717/peerj.12476_

## Round 0.1 · original submission · Minor Revisions

Review
I apologise for the delay in completing my review of your manuscript. My departure date for a planned short vacation arrived before I was able to complete my review.

Overview
This manuscript examined nesting habitat of Woodlarks in Upper Austria by comparing the habitat within 150 m of the territory center or nest site of 18 males with the habitat in 16 randomly selected non-territory sites. Of 56 variables estimated, nest sites were most closely associated with greater heterogeneity (Simpson’s diversity index), greater distance from dirt roads, greater spatial separation (disaggregation) of patches of bare soil, and lower patch density (which implied larger patches). Habitat heterogeneity had been identified as important in other regions. Bare ground was recognized as important, but apparently aggregation of patches and distance from dirt roads had not been examined. Unlike previous studies, vegetation height was not a significant variable, possibly due to differences in spatial or temporal scale between studies. In addition, the authors examined whether land use intensity changed over a 9-yr period (2007-2016), finding no significant change. The authors conclude that some characteristics of habitat can be generalized between regions but that others need to be identified at the local level.

The reviewers’ recommendations differed greatly. Reviewer 1 recommends rejection because of the limited sample size. The reviewer does not identify which conclusions are not valid due to the small sample size but indicates that the manuscript would be publishable in a ‘lesser’ journal. Reviewer 2 indicates that the manuscript needs only minor revisions to be publishable and that the methods and statistical analysis are appropriate. Because PeerJ policy is to publish valid scientific studies without regard to their impact, the manuscript is potentially acceptable, provided that the sample size was adequate for the questions asked. It seemed to me that there were strong, statistically significant findings and that few conclusions were based on findings of no effect, where the limited sample size could create potential problems. Therefore, I am prepared to consider a revised version for publication. My own review found numerous issues of clarity, need for additional information or discussion and grammar. Despite the number, my decision is ‘minor revisions’ because most require relatively small changes.

As someone with experience in habitat selection research, although not in birds, I found some of the Methods confusing or incomplete. Also, it was sometimes difficult to understand how the written description of some findings reflected the numerical data. In addition, there were inconsistencies in the organization of the manuscript and many small grammatical errors. Correcting these will make the manuscript easier for more readers to follow. Below, I provide specific comments on points that I found needed clarification. You may treat these as if they were a third review: make changes if the comments are valid or explain why you do not think that they need to be changed. I have also provided an annotated pdf with highlights and inserted comments to note minor errors of spelling, punctuation, grammar or formatting. You do not have to include the pdf comments in your rebuttal document unless you disagree and are not making the suggested changes.

Editor’s Comments
Some broader issues to consider
1) Literature review of Woodlark habitat associations (L63-73)
This paragraph is well written but not sufficiently linked to the present study. The stated objective of the manuscript is to assess whether Woodlark habitat use is similar in Upper Austria to other studied areas. However, the focus of this paragraph is on variation in habitat use over different spatial scales. The reader is not informed where the previous studies were carried out, how this relates to the species range, or what broad habitat type was investigated. Such information is needed to examine the question of variation in habitat use between different regions. Indeed, I wonder whether the underlying issue is really variation among regions or among habitat types (vineyards, orchards, pastures). Please consider the focus of the comparisons you can make and be sure that the Introduction introduces these concepts explicitly, including previous literature.

Clearly, variation in habitat use at different spatial scales is a complication for your objectives. Thus, there should be an emphasis also on studies that were carried out at similar scales. The relevance of this scale should be made clear so that your choice of a circle of 150 m radius makes sense to readers. (You indicate that this approximates territory size but not why other studies use much smaller areas.) The background should be clear enough that when you come to discuss your results readers can clearly see which and how many habitat attributes are similar and different. It would be reasonable, however, to be relatively brief in the Introduction (while still remaining focused on key issues) and go into more detail in the Discussion.

Given the focus on geographical variation in habitat use, the paragraph raising this issue (L74-83) seems superficial. The only reference provided (Whittingham et al. 2007) is 14 years old. What is the evidence that habitat use can change and by how much can it change? Is it really geographical variation or habitat type (e.g., vineyard vs. pasture)? Could some differences between studies by due to differences in methods, such as spatial scale or precision of habitat measures, or are does habitat use definitely vary regionally? What is known about possible causes or correlates of such changes or lack of changes? See also my specific comments below on the lack of clarity of individual sentences.

2) Historical changes in land use
Although you mention agricultural intensification as a probable cause of declines in farmland birds in the first paragraph, your objectives focus on habitat use in Upper Austria in comparison to other regions or habitats rather than the effect of intensification or changes over time. Therefore, I agree with Reviewer 2 that the examination of intensification does not seem to be relevant to this manuscript and should be removed unless there is a much more specific link between the two topics.

3) Discussion
For each of the habitat correlates you found, you need to expand the discussion. Please try to address the strength and reliability of the effect. How robust is the pattern? Are there possible confounding causes? How robust is the interpretation of the ecological patterns underlying the indices? After that, you can proceed to putting the result in the context of previous literature and what it means for the bird and for conservation efforts. If the use of variables related to landscape configuration is original to your study, you should indicate that and discuss whether such variables would be useful in future studies of this and other species. Since your goal is to discuss regional variation in habitat correlation, I would have expected some discussion of why some characteristics vary and others do not and of how to discover which are which.

More specific suggestions
Abstract
L19,30. Some sentences were too long and complex. I have suggested ways to shorten them.
L23-25. Is it correct that you mapped the territories? I thought that you estimated the territory center from activity or nest location and then assessed habitat within 150 m. This does not seem to be the same as mapping the territory.
L24. Why are the 16 non-territory sites not mentioned? Are they not integral to the analysis?
L24-25. It is not clear that ‘local landscape’ refers to habitat attributes referred to above. Why change the term?
L25-27. This statement of main results is not sufficiently clear and may be misleading. Revise after considering my comments below on the Results.
L32. My understanding from the figure and results is that probability of a territory increases with distance from roads, so this advice seems counterproductive.

Introduction
L34ff. I have provided suggestions to reduce the redundancy and wordiness of many sentences that follow.
L49. Give readers an idea of the geographical range.
L61. Not clear what you mean by ‘at the local landscape’. Are you referring to Upper Austria or to the spatial scale?
L65. Clarify the variables. Does tall, dense vegetation refer to clumps of grass, shrubs, or forest, all of which could be considered ‘tall’ depending on what they are compared to?
L69. Clarify ‘habitat amount’. It seems to me that all the space around a nest could be potential habitat, so all would have the same amount of habitat. Are you referring to the amount (or proportion) of one or more specific habitat types?
L76-79. This sentence is not clear, too long and has errors in prepositions. Do you mean something like ‘If habitat preferences differ among geographical areas, effective conservation requires management strategies that also vary locally’?
L79-82. It is not clear what you mean by this sentence. How does land use relate to the previous argument? It seems to me that over a broad geographical range, crops grown will relate to climate and soils which you do not mention. How do you expect farmers’ perspectives to change over geographical areas? Are you referring to cultural differences among regions or simply the decisions of individual landowners?
L85-86. Revise this sentence. It is not clear if you mean that vineyards are a habitat characteristic important in other areas or if you mean that vineyards are a different habitat type in which specific characteristics might change. The problem is partly grammatical and partly that you are referring to habitat at different scales in the same sentence, without being specific about what you mean. Move the habitat type (cropland-grassland-forest mosaic) up to L85 after the Fahrig reference to make it clearer that this is your focus. This would be the place to add Upper Austria as the specific focus of your study (cropland-grassland-forest mosaic in Upper Austria). Note that for the reader to understand this objective, you need to specify habitat types previously studied in the paragraph on habitat variation (L63ff).
L88. Clarify what you mean by habitat amount.
L88. Except for one sentence on fragmentation, you have not provided any introduction in the previous paragraphs for the reader to understand why you would focus on ‘spatial arrangement’ or indeed what this term means. ‘Configuration’ is synonymous but does not help the reader see precisely what you mean or why it is important.

Methods
L105. I don’t know what a ‘landscape element’ is. Is this an accepted term? By ‘tree’, do you mean individual or isolated trees to distinguish them from ‘groves’ (clusters of trees)? Or does ‘grove’ refer to orchards?
Fig. 1. This figure and the caption both need clarification.
• Does the map show the entire area of Muhlviertel or only part of it?
• Are there borders for the areas labelled Neumarkt and Nature Park? It is not clear why the Nature Park label is so far from the points.
• In the inset map, Vienna is shown as a dot that seems similar to ‘presence’, and in Upper Austria there appear to be three dots even though there are only two study sites.
• Latitude and longitude indicators would be useful, at least for the inset map of Austria.
• Is ‘built up’ an accepted term in landscape ecology? Does it refer only to buildings or to adjacent lawns and gardens? Is it correct as implied by the map that there are no built up areas (farms, storage buildings, houses) in the cropland? Some of these questions may reflect my lack of familiarity with terrestrial habitat terminology, but I want to be sure that all terms will be unambiguous to all readers in the field.
• Captions should provide a complete description of the figure.
L109ff. I found the description of territory mapping and habitat characterization to be confusing.
• I presume that the reason for ‘territory mapping’ was to estimate the territory center as the point around which habitat would be measured, but I am not sure that this is how readers will understand the term ‘mapping’.
• In addition, the paragraph refers to mapping absence areas. I am not sure how this could be done according to the described methods because there was no Woodlark to observe. If you used centers of occupied habitats and random points as the basis for habitat measures, then it seems that the description of random points should come in the next paragraph. If you somehow mapped absence areas separately from assessing the surrounding habitat, an explanation of the process and goal is needed.
• How many presence positions were based on the nest location? Is the nest typically near the center of the territory?
• How did you assign GPS coordinates to the bird positions? I assume that you were observing from a distance rather than right where the bird was, so you need to state how you determined the location and the error in this estimation.
• How did you select the random, absence locations? This is critical to the analysis because any biases in choice of the absence locations could strongly affect the results.
• Is the 266 m distance between presence and absence locations (L121) based on the centers or the circumference? If based on the centers, 150 m radii could potentially overlap.
• What do you mean by the mapped study area (L119)? This concept has not been explicitly introduced.
L123ff. The section on habitat characteristics also needs clarification. Remember that an independent researcher should be able to repeat the study.
• L128-129. Should there be a table, perhaps in supplementary material, with operational definitions of all habitat categories?
• How would a researcher trying to repeat the study distinguish grassland from rough pasture?
• What are groves?
• Is this a complete list of habitats, never roads, buildings, hedgerows, or water bodies, for example? Later, you mention roads and residential areas but do not indicate how they fit into the habitat characterization.
• L130-131. Do these measuring points refer only to vegetation height and DBH or were there other variables measured at these points? How did you measure height?
• L130. The sentence implies that the number of points was random, but then you explain a protocol that is not random. Do you mean that the location of the points was random? If so, what was the randomization procedure? If not truly random, you should use the term ‘haphazard’. In the latter case, you Discussion should consider the potential effect of bias in the haphazard selection of absence points.
• L133. What habitat was assigned to the small patches that were not characterized? Were they left out or included as part of adjacent larger patches?
Some of the analysis needs clarification.
• L138. Are PD, LSI, PROX, CONTAG and SIDI measures obtained using FRAGSTATS? If so, you need to make it explicit. You could revise to state ‘Using FRAGSTATS, we determined . . .’. The statement ‘we calculated’ suggests that these may have been additional measures. If that is the case, you need to specify the measures derived from FRAGSTATS and add an ‘also’ to ‘we calculated’.
• L140. Is this the density of patches of all types combined? How is a patch defined?
• L142. It is not clear why a measure of disaggregation is called shape. How is it measured? Is it a measure of aggregation of all patch types together or related to individual habitat type patches?
• L143. Are you sure that disaggregation is the correct term here. The dictionary definition of disaggregation is to divide into component parts. Do you mean dispersion, i.e. the degree to which patches are spaced out or aggregated?
• L144-145. Similar questions apply to the next two measures.
• LSI, PROX and CONTAG seem to be all measuring the same thing. Why did you choose these different measures? How do they differ?
• L147. After defining an acronym/abbreviation, you should use it consistently. If you do not need it, don’t use the abbreviation as it creates another barrier for the reader.
L157-159. The category ‘other’ is not clearly defined. It seems to include grasslands and other areas that could have been included in the above crop or woody vegetation.
L162. I don’t see how you got 56 variables. I think you need a table to list them all.
L176. I did not see a measure that would give you percentage of dirt roads.
L187. Does this ANOVA take into account the lack of independence between treatments (years)? I am not particularly sophisticated in my understanding of statistics, but this seems to be an important issue.

Results
L192. For ease of understanding by readers, it is important to use a consistent order of information throughout the manuscript. Because you presented landscape use changes last in Introduction and Methods, it should come last in Results and Discussion (if not eliminated as suggested above).
L196. Although it is not completely clear, you seem to be saying that fluctuations in land use were less than 5% between years. However, this does not seem to agree with Fig. 2 which shows a more substantial increase in wood and decrease in crop.
L198-199. The Methods did not prepare me for the measure ‘proportion of dirt roads’ and measures of bare soil or for a distinction between ‘patch density of the landscape’ and ‘patch density of grassland’ and the use of 20 cm to distinguish short and tall vegetated cropland. Please be sure that you Methods are clear and complete so that readers can readily understand the Results.
L208. Wouldn’t it be clearer to present the percentages for the same variables in the control/random sites?
L208-210. These are key results, so it is important to describe them accurately and in proper terms. Use the terms defined in the Methods, with additional words to explain concepts that are not intuitive or widely used.
• It seems to me that 50% is an arbitrary description of a sigmoid curve. The rate of increase is increasing before that and it is decreasing afterwards.
• It seems to me that the increase in probability of a territory rises with the degree of dispersion of bare soil patches, not the number of dispersed patches as your sentence seems to imply.
• What are the units of Fig. 4C? Patch densities seem much higher than I would have expected (but I do not know this area or Woodlark habitat).
• This statement is grammatically incorrect, and I don’t understand what you mean well enough to suggest a correction. The line seems to increase more or less continuously. I do not detect a threshold at 40 m.

Discussion
L214. Heterogeneity as a common factor deserves more attention. Are the components of heterogeneity similar among the different studies? How important a variable is it? How diverse are the habitats and geographic range over which this pattern has been found? Presumably, some combinations of habitats could be more important than others, even if they have the same level of heterogeneity.
L221-222. This sentence is not clear. What do you mean by ‘contribute to the patchiness’. You could write ‘contribute to prey availability and refuges from predation’. However, you need to provide some detail about the processes involved, with references, if possible. What do Woodlarks eat? In what microhabitat? Do they use cover to escape predators?
L225-227. This is not clear. You have not provided insight into how patch density is a measure of patch aggregation. If patch density is measured on the scale of the site, this would not be correct. I can see a possible negative relationship between patch density and patch size, but only if the amount of habitat in the patches is held constant.
L230ff. The previous paragraph makes the case for aggregated patches, but this paragraph seems to emphasize the opposite. I think more discussion on what type of patches increase probability of a territory with aggregation and which type of patches decrease it with aggregation.
L233-235. This may explain the importance of bare patches as habitat, but your variable is the dispersion of bare patches, and you have not offered an explanation for that.
L236. This paragraph highlights the importance of dirt roads as providing bare patches. However, your data show that territories increase with distance from dirt road, which seems opposite what you are saying.
L246ff. You need to explain previous work in more detail. What heights were relevant? If it was short height, then increasing vegetation height later in the season cannot explain your findings (unless the territories actually shift to new locations).
L264. It is still not clear to me how farmers’ perspectives relate to scale of estimation of change in land use.
L266. How is fertilizer more relevant and what is it relevant to?
L279. You have not justified a threshold of 40 m. It is not clear from the figure. Furthermore, this implies that the territory locations are known before creating a dirt road. Is this a practical conservation measure?
L280. I do not see a justification for the assertion that conservation of habitats that favor Woodlarks will also favor other farmland species. What is the case that other species need similar habitat elements?

References
Most of your references use abbreviated journal titles, but this is not consistent.
Most of your references use lower case for article titles in journals, but this is not consistent.

Table 1. The heading should indicate the independent and dependent variables that are examined with this GEE.
Fig. 3. Caption is incomplete. Figures with captions should be able to be understood on their own without detailed examination of the text.

Reviewer 1 ·

Basic reporting

See below.

Experimental design

See below.

Validity of the findings

See below.

Additional comments

General comments
Overall, the manuscript is well-written. Additionally, analyses on the habitat preferences of the woodlark are highly relevant for successful conservation management. However, my main concern with this study is that the sample size is clearly too low (18 territories vs. 16 non-territories) for a high-ranking journal such as Peer J with a broad international audience. Therefore, I would recommend submission to a more specialised ornithology journal such as Bird Study.

Specific comments
Title. Please delete ‘in Austria’ to address a wider readership.
Line 36. “An important driver”
Line 41. Please delete “and wetlands”; wetlands are unimportant for the study organism.
Line 97. “, Austria (Central Europe)”
Line 100. “We selected these areas since they host the main populations of the Woodlark in Upper Austria”
Line 192. “differ”

Reviewer 2 ·

Basic reporting

The topic is not new and the article does not add particularly informations on what we already know about this species; however, Lullula arborea is a specie of conservation concern and, above all, is a good indicator of the presence of diversified landscapes that, in this particular historical period, are threatened by a lot of different pressures and better understanding their ecological importance and functioning is very important

The article is clear, the reading is fluent and the hypotheses cited in the introduction are fully developed along the manuscript, even if the part regarding the land use changes are not fully deepened and it seems to be "external" from the rest of the article. The authors find no apparent links between land use changes and the decrease of Lullula arborea population, however in my opinion a more comphrensive discussion on which reasons could be at the base of it decline lacks. Written in this way this part does not have much sense, probebly it could be better to cut it at all.

The English language is properly used.

Figures and Tables are well structured and easily understandable; the references are complete and updated.

Experimental design

No comment

Validity of the findings

Except for the part related to the potential impacts of land use changes on the species population, which appears to be "external" from the rest of the article and does not add any relevant information, the initial hypotheses are clear and well developed along the manuscript.
Methods are clear and complete; one only thing: in line 112 you should give more details about the number of field surveys (how many?). Statistical analysis is appropriate and correctly used.
For what concerns the findings, these are well and comprehensively presented and seem to be consistent with what we already know about the ecology of the species. The approach to consider different (three) spatial scales gives the findings more interest.
One of the parameters which seem to have a positive effect on the presence of breeding territories of the species are the dirty roads; the authors postulate that it is because dirty roads are patches of bare soils, used by the species for accessibility of food resources. I agree with this statement, but I think you should analyze and exclude the possibility that this positive effect is not linked with the fact that where there are roads, the surveys are more easy, the probability to cover all the study area increases as well as, consequentely, the possibility to detect the species.

Additional comments

I think this article, with minor revisions and few more details, could be interest to better understand the ecological functionality and importance of cultural micro- and macro-diverse landscapes.

---

## Round 0.2 · Minor Revisions

Thank you for your careful revision and detailed explanation. It was relatively easy to examine the changes. I agreed with all but one of the changes.

L140. The selection of random sites is critical to the study. My comment asked how these locations were decided. You did not describe the procedure but only said that they were random. It is necessary to describe the algorithm that you used because this could critically affect the comparison with occupied nest sites. I suggest that you email me the revised wording (donald.kramer@mcgill.ca) before resubmitting so that I can confirm the change and thus provide final acceptance when you resubmit.

Also, please provide a key to all abbreviations on the data table in supplementary material. This would probably be clearest as a separate sheet labelled ‘Abbreviations’.

Finally, there are still a few minor grammatical problems. I have provided a pdf with corrections highlighted. If you do not agree with any of them, please email me to check so that the next resubmission will be correct.

---

## Round 0.3 · accepted · Accept

I consider the manuscript now ready for publication. Thank you for your efforts.